# Change in PD-L1 and CD8 Expression after Chemoradiotherapy for Esophageal Squamous Cell Carcinoma

**DOI:** 10.3390/biomedicines10081888

**Published:** 2022-08-04

**Authors:** Wei-Chung Chen, Chun-Chieh Wu, Yi-Hsun Chen, Jui-Ying Lee, Yao-Kuang Wang, Nian-Siou Wu, Ming-Tsang Wu, I-Chen Wu

**Affiliations:** 1Ph.D. Program in Environmental and Occupational Medicine, Kaohsiung Medical University, Kaohsiung 807, Taiwan; 2Research Center for Environmental Medicine, Kaohsiung Medical University, Kaohsiung 807, Taiwan; 3Department of Pathology, Kaohsiung Medical University Hospital, Kaohsiung Medical University, Kaohsiung 807, Taiwan; 4Division of Gastroenterology, Department of Internal Medicine, Kaohsiung Medical University Hospital, Kaohsiung 807, Taiwan; 5Division of Thoracic Surgery, Department of Surgery, Kaohsiung Medical University Hospital, Kaohsiung Medical University, Kaohsiung 807, Taiwan; 6Faculty of Medicine, Department of Medicine, College of Medicine, Kaohsiung Medical University, Kaohsiung 807, Taiwan; 7School of Medicine, College of Medicine, Kaohsiung Medical University, Kaohsiung 807, Taiwan; 8Department of Family Medicine, Kaohsiung Medical University Hospital, Kaohsiung Medical University, Kaohsiung 807, Taiwan

**Keywords:** esophageal squamous cell carcinoma, PD-L1, concurrent chemoradiation therapy, CD8, prognosis

## Abstract

Esophageal cancer has a dismal prognosis with a five-year survival rate below 20%. Recently, immunotherapy has become a new standard of care for this cancer; therefore, we aimed to examine the programmed death ligand 1 (PD-L1) expression in esophageal squamous cell carcinoma (ESCC) tissues before and after concurrent chemoradiation therapy (CCRT). In total, 64 patients with pre-CCRT ESCC specimens were examined for PD-L1 expression, with twenty-three of them having a partial response (N = 23) or stable disease (N = 1) after CCRT while post-CCRT tissue specimens were collected. All of them were tested for PD-L1 and 15 of them also had CD8 expression in the paired ESCC samples. The prevalence of PD-L1 positivity was 54.7% and we found a trend of decreased PD-L1 expression and increased CD8 positive signal after CCRT. High pre-CCRT PD-L1 H-score in tumors was related to poor prognosis (adjusted hazard ratio = 2.81; *p* = 0.02), although CD8 signal was not associated with overall survival either in pre- or post-CCRT treatment. In conclusion, we found that PD-L1 expression tended to decrease in CCRT responders and our result supports PD-L1 expression in tumor as a predictor of ESCC prognosis.

## 1. Introduction

After the first immune checkpoint inhibitor was approved by the U.S. Food and Drug Administration for melanoma in 2011, its applications on other cancer types have bloomed and have been recently approved for esophageal and gastroesophageal junction cancers [1,2]. Eight published clinical trials and more than twenty-eight ongoing trials have targeted programmed death 1 (PD-1) or programmed death ligand 1 (PD-L1) alone or in combination with chemotherapy or radiotherapy to seek better responses to esophageal squamous cell carcinoma (ESCC) [3,4]. Two drugs targeting PD-1 (nivolumab and pembrolizumab) have shown better overall survival (OS) (10.9 vs. 8.4 months and 8.2 vs. 7.1 months) than chemotherapy [5,6] or better disease-free survival (DFS) against placebo (29.7 months vs. 11 months) [7] in esophageal cancer.

The prevalence of PD-L1 positivity has ranged from 18.4% to 80.8% in ESCC and its relationship with ESCC prognosis is still controversial [3,8,9,10,11]. Some studies have reported positive expression of PD-L1 as being correlated with poor OS (*p* = 0.010) and DFS (hazard ratio (HR) = 1.436; *p* = 0.009) [8,9], while another study has shown the opposite results where high PD-L1 was related to a favorable prognosis with adjuvant radiotherapy (84.4 months vs. 36.0 months; *p* = 0.046) [11]. A meta-analysis published in 2016, including 1350 ESCC patients, found no association between clinical characteristics and PD-L1 expression, but there was a trend of worse prognosis in patients with higher PD-L1 expression (HR = 1.65; 95% CI = 0.95–2.85; *p* = 0.07) [12]. Another meta-analysis of 2877 ESCC patients indicated high PD-L1 expression was associated with poor OS (HR = 1.38; 95% CI = 1.02–1.86; *p* = 0.04), especially in Asian ethnicities (HR = 1.49; 95% CI = 1.11–1.99; *p* = 0.008) [13], although, a more recent meta-analysis of 3677 ESCC patients showed PD-L1 expression was neither correlated with OS (HR = 1.16; 95% CI = 0.94–1.42; *p* = 0.16) nor DFS (HR = 0.85; 95% CI = 0.66–1.10; *p* = 0.21) [14].

Cytotoxic CD8-positive T cells are the most powerful effectors in the anticancer immune response. The prevalence of CD8 positivity in ESCC tissues ranges from 2.39% to 57.5% [15,16,17], but its relationship with overall survival appears equivocal with negative findings in most studies [9,18,19,20]; moreover, few studies have reported a change of PD-L1 expression before and after CCRT, nor its relationship with CD8 T cell expression or patients’ outcome. Thus, we aimed to investigate the alteration of PD-L1 expression in paired pre- and post-CCRT specimens to examine its correlation with clinical outcomes. We also collected more ESCC specimens to reconfirm the prevalence of pre-treatment PD-L1 expression in Southern Taiwan.

## 2. Materials and Methods

### 2.1. Patients and Specimens

This is a retrospective cohort study. In total, 86 ESCC patients received either direct esophagectomy or neoadjuvant concurrent chemoradiotherapy (CCRT) and surgery at Kaohsiung Medical University Hospital (KMUH) between 2011 and 2017, with 64 being eligible after fulfilling the following criteria: (i) being older than 20 years; (ii) having pathologically confirmed squamous cell carcinoma and with sufficiently archived tissue blocks for subsequent immunohistochemistry staining (IHC); and (iii) having survived more than three months after diagnosis. Clinical and pathological information was obtained from the review of medical records (Figure 1). This study protocol was approved by the Institutional Review Board of KMUH (KMUHIRB-E(II)-20180261) and written informed consents were obtained from all the patients.

### 2.2. Immunohistochemistry Staining for PD-L1 and CD8

All the tissue specimens were preserved in formalin-fixed paraffin-embedded (FFPE) form. After reviewing the hematoxylin and eosin staining slide from the archived tissue block, the representative tissue blocks were selected and sectioned to 4 μm thickness for staining. First, the specimens were deparaffinized through serial baths in xylene and rehydrated in a series of diluted alcohol and then the antigen was retrieved with target retrieval solution (Target Retrieval Solution, Citrate pH 6 (10×), Cat no. S2369, DAKO Agilent, Santa Clara, CA, USA) by autoclaving. After removal, endogenous peroxidase activity and nonspecific background staining were performed by mounting 3% hydrogen peroxide for 5 min at room temperature.

We used two different PD-L1 antibodies (PD-L1 clone E1L3N antibody and PD-L1 clone 22C3 antibody) to stain cancer tissue specimens in different cancer patients in this study. First, anti-PD-L1 E1L3N antibody (Cat no. #13684, Cell Signaling Technology, Danvers, MA, USA) was mounted and incubated on specimens with coverslip at room temperature following TBS washing in triplicate. The diluting ratio of PD-L1 primary antibodies was 1:100. Two hours later, the slides were rinsed with TBS twice and secondary antibody was applied and then conjugated with horseradish peroxidase (HRP) (ChemMate™ DAKO EnVision™ Detection Kit, Cat no. K5007, DAKO Agilent, Santa Clara, CA, USA) for 30 min of incubation. Finally, the slides were washed with TBS twice and visualized by incubating with 0.03% 3,3V-diaminobenzidine tetrahydrochloride (ChemMate™ DAKO EnVision™ Detection Kit, Cat no. K5007, DAKO Agilent, Santa Clara, CA, USA) for 5 min. All slides were counterstained with Mayer’s hematoxylin for 30 s, and sealed with a non-aqueous mounting medium (Entellan^®^, Cat no.1.03961, Merck, Darmstadt, Germany) after dehydration. For the PD-L1 22C3 antibody (Cat no.SK006, DAKO Agilent, Santa Clara, CA, USA), staining is regularly used in our Department of Clinical Pathology to identify and choose suitable cancer patients for immunotherapy. We used the automated staining mechanism BenchMark ULTRA system and followed the manufacturer’s instructions (https://www.agilent.com/cs/library/usermanuals/public/29349_22c3_pharmdx_nsclc_interpretation_manual_kn042.pdf accessed on 5 December 2021). The CD8 staining followed the same protocol as for the use of anti-PD-L1 E1L3N. The primary CD8 antibody (Cat. Ab4055, Abcam, Cambridge, UK) was diluted by 1:1000.

### 2.3. Evaluation of PD-L1 and CD8

The results of the IHC stain were evaluated by one pathologist (Dr. Chun-Chieh Wu), who was blinded to the patients’ clinical conditions and survival. Each tissue slide must contain more than one hundred tumor cells, which was also qualified as an adequate residual tumor in the post-CCRT group. The PD-L1 signal intensity on tumor membrane was scored into four grades (Appendix A): Score 0: no appreciable 3,3V-diaminobenzidine tetrahydrochloride (DAB) signal above background; Score 1: weak positive intensity tumor membrane staining; Score 2: moderate positive intensity tumor membrane staining; and Score 3: strong positive intensity tumor membrane staining. In addition to intensity score, the percentages of stained tumor cell samples were also recorded. H-score was also calculated based on the PD-L1 signal intensity score and staining percentages and the same for both PD-L1 22C3 and E1L3N, which were
H-score = (percentage of score 3) × 3 + (percentage of score 2) × 2 + (percentage of score 1) × 1

The range of H-score can be from 0 to 300 [11].

The numbers of CD8+ T cells were calculated by Image J under five high-power magnification field (400 ×) images randomly selected from each slide. Each image must contain at least 100 tumor cells and a stroma region in the same field. The counting of CD8 signals was performed as described in the reference cited [21]. In brief, we deconvoluted the color of DAB to extract the candidate CD8 signals and used the Otsu algorithm as a cutoff to quantitively determine the signals, followed by the watershade function to split the connective signals, and then an area of reference for CD8 signal was used as a cutoff to count CD8 positive signals.

### 2.4. Quality Control for PD-L1 of E1L3N Clone

The Hodgkin lymphoma HDLM2 and prostate cancer (PC3) cells were used as positive and negative controls to evaluate the E1L3N clone specificity (Appendix A) [22]. Due to insufficient archive tissues, the PD-L1 E1L3N antibody was re-stained in 28 patients’ cancer tissues samples that had the data of PD-L1 22C3 staining, including 3 patients with both pre- and post-CCRT tissues and 4 patients with pre-CCRT tissue only in the CCRT paired group, 20 patients with pre-CCRT in the pre-CCRT group, and 1 patient in the surgery only group (Figure 1). There was a good correlation of PD-L1 expression using these two antibodies counted either by H-score or stained tumor percentage (Spearman correlation r = 0.50, *p* = 0.007 and r = 0.54, *p* = 0.003, respectively) (Appendix A).

### 2.5. Statistical Analysis

The Spearman test was used to analyze the correlation between two anti-PD-L1 antibody clones, while the Wilcoxon signed-rank test was applied to calculate the significance of PD-L1 alteration after receiving CCRT. Then, the duration of the overall survival rate by PD-L1 H-score was analyzed by the Kaplan–Meier method. Cox proportional hazards regression was used to validate the effect of PD-L1 H-score on the duration of overall survival after adjusting for other potential confounders. These potential covariates included age (treated as continuous), gender (women vs. men), clinical stages (early stages I and II vs. late stages III and IV) and treatment (received endoscopic submucosal dissection or surgical resection vs. received concurrent chemoradiotherapy). When the data of both anti-PD-L1 22C3 and anti-PD-L1 E1L3N in the same patient were available, the former was used to conduct the statistical analysis performed using IBM SPSS version 20.0 software. All *p*-values were two-sided and statistical significance was defined as *p*-value < 0.05.

## 3. Results

### 3.1. Study Subjects

Among the 64 eligible ESCC patients, 45 received CCRT and 20 of them further received esophagectomy (Figure 1). Twenty-three patients had paired specimens from endoscopic biopsies before CCRT and biopsy or resected tissues after CCRT. Of these 23 patients, 3 patients were evaluated with both anti-PD-L1 ElL3N and anti-PD-L1 22C3, 15 patients with anti-PD-L1 ElL3N only and 5 patients with anti-PD-L1 22C3 only. The remaining 22 of the 45 patients only had specimens from pre-treatment endoscopic biopsies. Of these 22 patients, 20 patients were evaluated with both anti-PD-L1 ElL3N and 22C3 and 2 patients with anti-PD-L1 22C3 only (Figure 1). For the 19 patients with surgery alone, one patient was evaluated with both anti-PD-L1 ElL3N and 22C3 and 18 patients with anti-PD-L1 ElL3N only.

The clinicopathological characteristics of all enrolled patients and those with paired specimens across CCRT are shown in Table 1 and Appendix A, respectively. The majority of subjects were male (95.3%), and the tumors were commonly moderately differentiated (76.5%) and at stage III (56.2%). Five patients received endoscopic submucosal dissection (ESD) and fourteen underwent direct esophagectomy (OP). Definite CCRT was delivered to 25 (39.1%) of the patients and 20 (31.2%) had neoadjuvant CCRT followed by surgery. Most patients (32/45 = 71.1%) had a partial response after CCRT, while one had progressive disease (Table 1). The baseline PD-L1 H-score ranged from 1 to 225 for all patients and these provided paired specimens across CCRT. We used the median level (H-score = 2) as the cutoff value for further analysis (Appendix A). The positive rate of PD-L1 (H-score ≥ 2) was 54.7% in all patients (Table 1) and was similar among those with and without paired specimens (52.2% and 56.1%, respectively; Appendix A).

### 3.2. Down Regulation of PD-L1 in ESCC after CCRT Treatment

Among the 23 patients with paired specimens, there was a non-significantly decreasing trend of PD-L1 H-score after CCRT (Wilcoxon signed-rank test, *p* = 0.135) (Figure 2A). A similar trend of PD-L1 down regulation was seen in the subgroup analysis using either the 22C3 antibody (N = 8) or the E1L3N antibody (N = 15) (Wilcoxon signed-rank test, *p* = 0.156 and 0.496, respectively) (Figure 2B,C). Moreover, we observed a good correlation of PD-L1 expression in the three patients with enough paired tissue specimens before and after CCRT to validate the two antibodies (Figure 2D).

### 3.3. Prognostic Relevance of Tumor PD-L1 Expression in ESCC Patients

The median value of the pre-CCRT PD-L1 H-score (median cut-point: 2, Appendix A) was used to dichotomize 64 patients into two groups to examine the prognosis. The Kaplan–Meier curve showed a higher pre-CCRT H-score of PD-L1 (≥median, 2) with significantly worse survival compared to a lower H-score than the median (Log rank test, *p* = 0.003) (Figure 3A). The same result was also noted in the subgroup of 23 patients with paired tissues across CCRT (Log rank test, *p* = 0.006) (Figure 3B).

After adjustment for covariates (gender, age, cancer stage and treatment strategy), patients with higher pre-CCRT PD-L1 had shorter overall survival than those without (HR = 2.814, 95% confidence interval CI = 1.196–6.618, *p* = 0.018) (Table 2). The results remained similar in subgroup analysis of 23 patients with paired specimens after adjustment for gender and age (HR = 3.46, 95% CI = 1.132–10.574, *p* = 0.029) (Table 3); however, survival analysis was borderline significant in subgroup analysis of the 23 patients with paired specimens across CCRT before (Log rank test, *p* = 0.065) or after adjustment for gender and sex (HR = 0.466, 95% CI = 0.163–1.133, *p* = 0.154) (Appendix A).

### 3.4. CD8 Positive Cells in Tumor Specimens before and after CCRT

CD8 density in 15 ESCC patients with paired pre and post-CCRT specimens (E1L3N group in Figure 1) was also examined. Among the 15 patients with paired specimens, there was a non-significantly increasing trend of CD8 positive after CCRT (Wilcoxon signed-rank test, *p* = 0.08) (Figure 2E). After dichotomizing by median of CD8 IHC results, no association was found between pre-CCRT CD8 level and patients’ survival (Log rank test, *p* = 0.268 in Figure 3C and HR = 0.66, 95% CI = 0.18–2.35, *p* = 0.52 in Table 4); however, there was borderline significance for those with lower post-CCRT CD8 level to have a longer survival period (Log rank test, *p* = 0.06, Figure 3D and HR = 4.89, 95% CI = 0.96–24.92, *p* = 0.06, Table 4). There were no obvious correlations between PD-L1 H-score and CD8 IHC results either in pre-CCRT (HR = 0.36, 95% CI = 0.03–3.95, *p* = 0.40) or post-CCRT group (HR = 1.32, 95% CI = 0.13–13.92, *p* = 0.82) after adjusting for gender and age (Appendix A).

## 4. Discussion

Neoadjuvant or definite CCRT is the common first-line treatment for advanced/unresectable ESCC [23]. Recently, anti-PD-1 blockades such as nivolumab and pembrolizumab have been approved by the US Food and Drug Administration

(FDA) is second-line or first-line treatment comparing or adding on traditional chemotherapy for patients with recurrent, locally advanced or metastatic esophageal cancer [2]. Patients with SCC or positive PD-L1 expression, defined as a combined positive score of 10 would enjoy better survival and response using pembrolizumab [6].

Many studies have investigated the expression rate and association between pre-treatment PD-L1 expression and ESCC patients’ outcomes (Appendix A). From these studies, the expression rate of PD-L1 in ESCC was 24.4–61.7% among Chinese, 18.9–63.3% among Japanese and 33.5–56.9% among Korean populations. Chen et al. found those with higher PD-L1 expression had worse outcomes after radiotherapy [11], although another study in Japan found no association between PD-L1 expression and chemotherapy response or survival [24]. One Korean study on 12 ESCC patients, who had pre- and post-CCRT tissue specimens, found PD-L1 expression increased after CCRT, but not after chemotherapy [25]. Their following study, which used whole exome sequencing and whole transcriptome sequencing on 29 ESCC patients with paired before and after CCRT samples, showed that both tumor mutation burden and neoantigen load reduced, and the PD-L1 protein expression had no definitive change [26]. The other Chinese study with 82 paired pre- and post-CCRT tissue specimens found a significantly increasing H-score after CCRT, but neither pre- nor post-CCRT PD-L1 expression was correlated with prognosis [20]. These controversial results are properly contributed by different anti-PD-L1 antibodies used in each publication. Though the U.S. FDA has approved three anti-PD-L1 antibody clones for clinical diagnosis in lung cancer, including 28-8, 22C3 and SP142 clones; the PD-L1 proportion score of SP142 was an outlier compared with 28-8, 22C3 and E1L3N clones on both tumor and immune cells in 90 lung cancer patients [27]. Interestingly, two hundred and sixty lung cancer patients assayed with high SP142 scores showed better object response rate and longer progression-free survival than those with high 28-8, 22C3 or SP263 scores after receiving immune checkpoint inhibitors [28].

For our 23 ESCC patients with pre- and post-CCRT tissue specimens, the H-score of PD-L1 in tumors decreased in 11 patients (47.82%) and increased in five patients (21.7%) after CCRT. The difference might come from having different ethnicities. Further, all 23 patients were CCRT responders (22 partial responses and 1 stable disease). Consistent with most of the previous studies, we found higher PD-L1 H-score in pre-CCRT tumor specimens was a significant predictor of poor outcome in all patients or subgroup analysis of the 23 patients with paired specimens, although post-CCRT tumor PD-L1 expression was not related to survival. We also found a trend of decreasing PD-L1 H-score and increasing cytotoxic CD8-positive T-cells in paired tumor specimens among CCRT responders.

Unfortunately, such beneficial immune change of reduced cytotoxic T-cell exhaustion in theory did not translate to survival time in our study. Our results imply the outcome after traditional therapy (resection or CCRT) was related to the initial tumor microenvironment rather than the immune status after CCRT. In addition, the Korean team showed neutrophils increased after CCRT in both complete response and non-complete response groups by using whole exome sequencing and whole transcriptome sequencing [26]. Together with our results, it suggests tumor immune microenvironment altered by radiation, chemotherapy or CCRT consists of multiple factors such as regulatory T-cell recruitment and activation during CCRT and the relationship between chemoradiation-resistant cell and PD-L1 expression.

There are several limitations to this study. Firstly, our sample size was restricted with limited paired pre- and post-CCRT tissue specimens. The pathological complete response rate after neoadjuvant CCRT was about 40% in our hospital and some patients refused surgery after CCRT, so it was not easy to obtain post-CCRT specimens. Secondly, although two anti-PD-L1 antibody clones showed modest to strong correlation, there were still some variations in PD-L1 staining percentages and intensity that might have led to false negatives. Thirdly, this was a single-center study and the results might differ from other hospitals using different dosages and regimens of CCRT. Regardless, this is the first study in Taiwan to provide regional data on the change of PD-L1 and CD8 expressions after receiving CCRT and its impact on survival.

In conclusion, our study showed a trend of decreasing PD-L1 expression and increasing CD8 signal after CCRT in responders. Higher expression of pre-treatment PD-L1 predicted a poor prognosis. Future studies are needed to clarify the details of the change in immune response and the mechanism behind it.

## Figures and Tables

**Figure 1 biomedicines-10-01888-f001:**
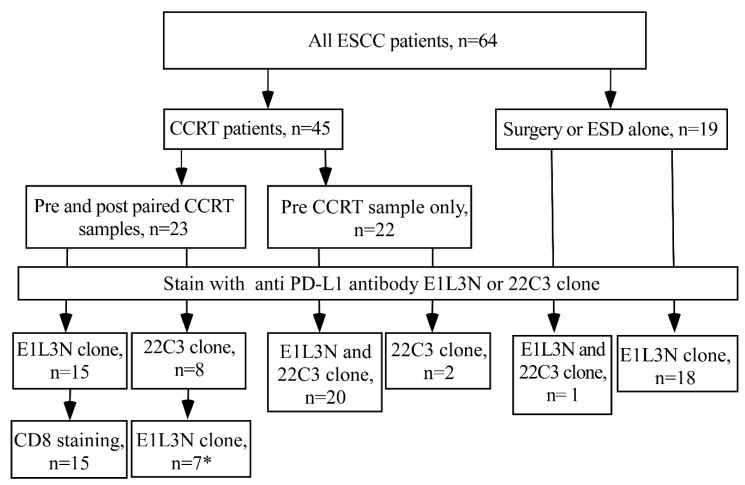
Study scheme and tissue staining by PD-L1 E1L3N clone antibody and/or PD-L1 22C3 clone antibody. CCRT, concurrent chemoradiation therapy; ESCC, esophageal squamous cell carcinoma; ESD, Endoscopic submucosal dissection * Due to insufficient tissue block, 3 out of 8 were stained on pre- and post-CCRT group, and 4 out of 8 were stained on pre-CCRT group.

**Figure 2 biomedicines-10-01888-f002:**
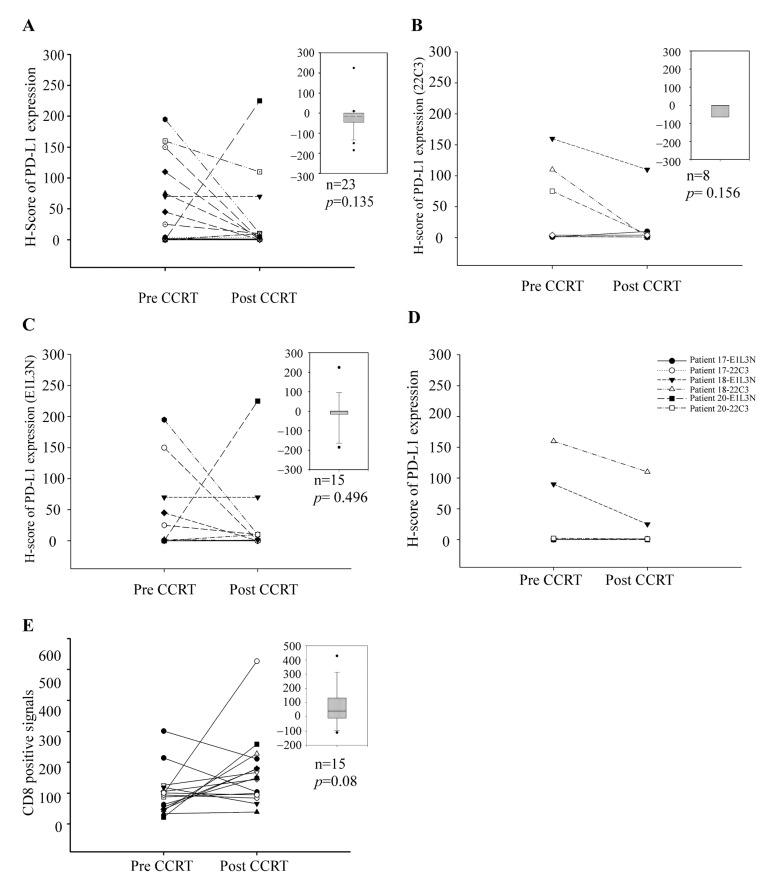
Decreased tumor PD-L1 H score after concurrent chemoradiotherapy treatment. (**A**) Quantification of PD-L1 expression before and after CCRT treatment in 23 paired patients. (**B**) Quantification of PD-L1 expression before and after CCRT treatment in the other 8 paired patients with anti-PD-L1 22C3 clone. The *p* value is 0.156 with Wilcoxon signed-rank test. (**C**) Quantification of PD-L1 expression before and after CCRT treatment in 15 paired patients with anti-PD-L1 E1L3N clone. The *p* value is 0.496 with Wilcoxon signed-rank test. (**D**) Quantification of PD-L1 expression before and after CCRT treatment in 3 paired patients with both anti-PD-L1 E1L3N and 22C3 clone. (**E**) Quantification of CD8 expression before and after CCRT treatment in 15 paired patients. The *p* value is 0.08 with Wilcoxon signed-rank test. Abbreviations: CCRT, concurrent chemoradiation therapy.

**Figure 3 biomedicines-10-01888-f003:**
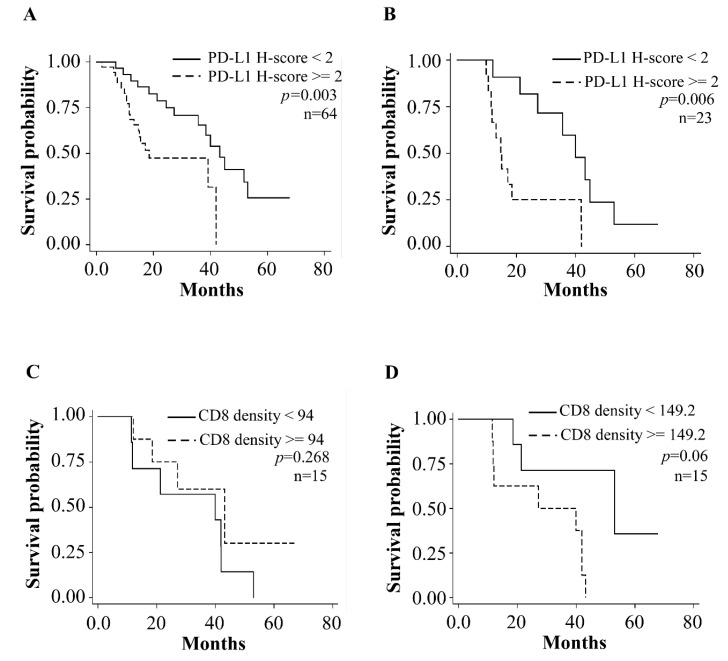
Kaplan–Meier survival curves dichotomized by the median of PD-L1 and CD8 H-score in ESCC specimens. (**A**) Patients with high pre-CCRT PD-L1 H-score had significantly worse overall survival in all patients (n = 64, *p* = 0.003); (**B**) in patients with paired specimens (n = 23, *p* = 0.006); (**C**) pre-CCRT CD-8 was not associated with patient survival (n = 15, *p* = 0.268); (**D**) patients with lower post-CCRT CD8 had borderline better survival (n = 15, *p* = 0.06).

**Table 1 biomedicines-10-01888-t001:** Clinicopathological characteristics of the 64 ESCC patients.

Characteristics	Mean ± SD or No. (%)	
**Gender**		
Male	61	(95.3)
Female	3	(4.6)
**Age (years) (Mean)**	56.08	±8.17
**Pathologic status**		
Stage I	10	(15.6)
Stage II	14	(21.8)
Stage III	36	(56.2)
Stage IV	4	(6.2)
**Tumor differentiation**		
Grade 1 (Well)	5	(7.8)
Grade 2 (Moderate)	49	(76.5)
Grade 3 (Poor)	6	(9.3)
Missing	4	(6.2)
**Treatment**		
ESD only	5	(7.8)
OP only	14	(21.9)
CCRT	25	(39.1)
CCRT then OP	20	(31.2)
**CCRT Response** **All CCRT cases (N = 45)**		
Complete response	7	(15.6)
Partial response	32	(71.1)
Stable disease	5	(11.1)
Progressive disease	1	(2.2)
**With paired specimens before and after CCRT (N = 23)**		
Partial response	22	(95.6)
Stable disease	1	(4.4)

Abbreviations: CCRT, Concurrent chemoradiation therapy; ESD, endoscopic submucosal dissection; OP, esophagectomy.

**Table 2 biomedicines-10-01888-t002:** Multivariate analysis of factors associated with overall survival of the 64 patients.

Variables	No.	Adjusted HR	95% CI	*p* Value
**Pre-CCRT PD-L1 H score**				
H-score Median <2	29	1		
H-score Median ≥2	35	2.81	1.20–6.62	0.02
**Gender**				
Male	61	1		
Female	3	0.72	0.09–5.55	0.76
**Age**	64	1.07	1.10–1.12	0.01
**Stage**				
Stage I and II	24	1		
Stage III and IV	40	2.34	0.67–8.16	0.18
**Treatment**				
With OP or ESD	19	1		
With CCRT	45	0.39	0.10–1.52	0.17

Abbreviations: HR, hazard ratio; CI: confidence interval; ESD, endoscopic submucosal dissection; OP, esophagectomy; CCRT, concurrent chemoradiation therapy.

**Table 3 biomedicines-10-01888-t003:** Multivariate analysis of factors associated with overall survival in the 23 ESCC patients with paired specimens before CCRT by median of PD L1 H-score.

	Variables	Total No. 23	PD-L1 Adjusted HR (95% CI)	*p* Value
Pre-CCRT	PD-L1 H score †			
	Lower than median	11	1	
	Equal and higher than median	12	3.46 (1.13–10.54)	0.03
	Gender			
	Male	22	1	
	Female	1	2.59 × 10^−17^ (0)	1.00
	Age	23	1.04 (0.98–1.11)	0.2
Post-CCRT	PD-L1 H score †			
	Lower than median	14	1	
	Equal and higher than median	9	1.31 (0.50–3.45)	0.58
	Gender			
	Male	22	1	
	Female	1	6.50 × 10^−17^ (0)	1.000
	Age	23	1.06 (0.99–1.14)	0.099

† The median value of PD-L1 H-score is 2.

**Table 4 biomedicines-10-01888-t004:** Multivariate analysis of factors associated with overall survival in the 15 ESCC patients with paired specimens before CCRT by median of CD8 density.

		Total No. 15	CD8 Adjusted HR (95% CI)	*p* Value
Pre-CCRT	CD8 density †			
	Lower than median	7	1	
	Equal and higher than median	8	0.66 (0.18–2.35)	0.52
	Gender			
	Male	14	1	
	Female	1	2.45 × 10^−17^ (0)	1.00
	Age	15	1.10 (0.99–1.21)	0.07
Post-CCRT	CD8 density ‡			
	Lower than median	7	1	
	Equal and higher than median	8	4.89 (0.96–24.92)	0.06
	Gender			
	Male	14	1	
	Female	1	2.20 × 10^−17^ (0)	1.00
	Age	15	1.11 (1.01–1.22)	0.03

† For pre-CCRT specimens, the median value of CD8 density is 94.2. ‡ For post-CCRT specimens, the median value of CD8 density is 149.2.

## Data Availability

Not applicable.

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
