# Peer review of "Change in PD-L1 and CD8 Expression after Chemoradiotherapy for Esophageal Squamous Cell Carcinoma"

_biomedicines, 2022, doi:10.3390/biomedicines10081888_

Round 1

Reviewer 1 Report

Chen et al., have studied the change in PD-L1 and CD8 expression in esophageal squamous cell carcinoma after chemoradiotherapy. They have shown a trend of decreasing PD-L1 expression and increasing CD8 signal after CCRT in responders and supports that higher expression of PD-L1 in pre-treatment groups, predicts a poor prognosis. This study is straightforward, supported by simple methods adopted, clear figures and appropriate statistical evaluation of the data. The data is nicely presented and the results are supporting the hypothesis. Although larger database is needed to further validate the findings. However, overall presentation of the data and English grammar and language needs improvement to make it better. I would suggest using some software to improve the quality of the language. Moreover, I would also suggest the authors to elaborate the introduction part with suitable references.

This study would be used by scientists since it provides adequate information and is very targeted, useful, and specific for clinician as well as basic researchers, who wish to study this disease. Therefore, I recommend the paper and this work should be accepted for publication with minor corrections suggested for language improvement and English phrases.

Author Response

Response to Reviewer 1 Comments

Point 1. However, overall presentation of the data and English grammar and language needs improvement to make it better. I would suggest using some software to improve the quality of the language. Moreover, I would also suggest the authors to elaborate the introduction part with suitable references.

Response: Thank you very much for your comments and very helpful suggestion. We elaborated the introduction section with results from published clinical trials and meta-analysis. Please check the updated Introduction from line 47 through line 67, and new references from line 380 through line 411.

Point 2. Therefore, I recommend the paper and this work should be accepted for publication with minor corrections suggested for language improvement and English phrases.

Response: This manuscript had been reviewed by a qualified English editor. Please see the certificate in the attached file.

Reviewer 2 Report

  1. The quality of each figures, should presented in vector format or use high resolution bitmap format.
  2. Line 236 "CD8 expression" is not clear, it is suggested to change into "CD8 positive cells" (or signals)
  3. It is suggested to provide a full detailed table for all patients' information inaddition of the descriptive summarization.
  4. It is interesting to clarify whether the downregulation of PD-L1 signal after CCRT is caused by globally reducing of tumor cells or merely downregulated PD-L1  expression in tumor cells?

Author Response

Response to Reviewer 2 Comments

Point 1. The quality of each figures, should presented in vector format or use high resolution bitmap format.

Response: Thank you for the comment. All the figures have been re-exported with 1000 pixels/inch in bitmap format. Please check revised figures.

Point 2. Line 236 "CD8 expression" is not clear, it is suggested to change into "CD8 positive cells" (or signals)

Response: We have revised “CD8 expression” in line 236 to “CD8 positive cells”. Thank you.

Point 3. It is suggested to provide a full detailed table for all patients' information in addition of the descriptive summarization.

Response: All patients information was summarized in Table S1. If Table S1 does not meet your standard, please let us know what kinds of patients’ information are needed.

Point 4. It is interesting to clarify whether the downregulation of PD-L1 signal after CCRT is caused by globally reducing of tumor cells or merely downregulated PD-L1 expression in tumor cells?

Response: Thank you for the comment. Although all the paired specimens were from patients with partial response, they were confirmed to have adequate residual tumors for evaluating PD-L1 staining percentage. We found along with globally tumor shrinkage, the percentage of PD-L1 expression also decreased in CCRT responders. We add a definition about adequate residual tumors from line 119 through 121 in Materials and Methods section as following: “Each tissue slide must contain more than one hundred tumor cells, which was also qualified as adequate residual tumor in Post-CCRT group.”